# Data Reproducibility and Effectiveness of Bronchodilators for Improving Physical Activity in COPD Patients

**DOI:** 10.3390/jcm9113497

**Published:** 2020-10-29

**Authors:** Yoshiaki Minakata, Seigo Sasaki

**Affiliations:** Department of Respiratory Medicine, National Hospital Organization Wakayama Hospital, Wakayama 644-0044, Japan; sasaki.seigo.su@mail.hosp.go.jp

**Keywords:** physical activity, pharmacological treatment, COPD, reproducibility, sedentary behavior

## Abstract

Increasing physical activity (PA) in patients with chronic obstructive pulmonary disease (COPD) is an important issue, however, the effect of bronchodilators on PA is still controversial. The indicators of PA, as measured by an accelerometer, can easily fluctuate based on several factors, which might cause inconsistent results. In this review, we listed the indicators of PA and the factors influencing the reproducibility of indicators of PA, and reviewed reports in which the effects of bronchodilators on PA were evaluated by an accelerometer. Then, we investigated the association between the processing of influencing factors and the effectiveness of bronchodilators for improving the PA of COPD patients. Fifteen reports were extracted using the PubMed database. In all seven reports in which adjustment was performed for at least two of four influencing factors (non-wear time, data from days with special behavior, environmental factors, and number of valid days required to obtain reproducible data), bronchodilators showed beneficial effects on PA. No adjustment was made for any of these factors in any of the four bronchodilator-ineffective reports. This suggests that the processing of influencing factors to secure reproducibility might affect the results regarding the effectiveness of bronchodilators for improving PA in COPD patients.

## 1. Introduction

Chronic obstructive pulmonary disease (COPD) is a disease characterized by airflow limitation and air trapping, which is usually caused by exposure to noxious particles or gases [1]. Air trapping leads to the sequential development of exertional dyspnea and decreased exercise capacity. Physical activity (PA) is reduced by decreased exercise capacity and other factors, such as psychological, socio-economic, comorbidities, and/or skeletal muscle weakness (Figure 1). This leads to a downward spiral of inactivity which predisposes patients to a reduced quality of life, and increased rates of hospitalization and mortality [2,3,4,5,6]. Reduced level of PA in COPD was associated with exacerbation [6,7,8,9,10], FEV1 decline [11,12,13,14,15], and mortality [4,5,6]. Furthermore, PA level was reported to be the strongest risk factor for all-cause mortality in COPD patients [5]. Therefore, the maintenance or improvement of PA becomes an important target for the management of COPD patients [1]. Recently, objective measurements using an accelerometer have been widely used to evaluate PA rather than questionnaires, which were mainly used but tended to overestimate the PA [16,17]. An accelerometer can detect acceleration in one, two, or three dimensions (uniaxial-, biaxial-, triaxial-accelerometer, respectively) and can calculate the values of several indicators of PA.

‘Physical activity’ can be defined as “any bodily movement produced by skeletal muscles that results in energy expenditure” [18], but it is usually defined as physically active behavior that is comparable to moderate to vigorous-intensity PA (MVPA) [19,20]. ‘Physical inactivity’ is defined as “a PA level that is not sufficient for meeting present PA recommendations” [20,21]. Physical inactivity in adults is often defined using threshold activity values: the non-achievement of 150 min of MVPA per week or 75 min of vigorous-intensity PA (VPA) per week or an equivalent combination of moderate- and vigorous-intensity activity [20]. In contrast, ‘sedentary behavior’ is defined as “any waking behavior characterized by an energy expenditure ≤1.5 metabolic equivalents (METs), while in a sitting, reclining or lying posture” [20,22,23] (Table 1).

Bronchodilators are key drugs in pharmacotherapy for COPD and have been reported to have beneficial effects on the pulmonary function, dyspnea, exercise capacity, and exacerbations in COPD patients [24,25,26,27,28,29,30,31,32,33,34,35,36,37,38]. However, its effectiveness in improving PA has differed among reports and is still controversial. The PA in COPD patients can be influenced by many factors; thus, it might not be improved by pharmacotherapy alone. On the other hand, the values of indicators of PA can easily fluctuate based on several factors, in other words, the indicators of PA show poor reproducibility. This might be the reason why the different studies have reported inconsistent results regarding the effects of bronchodilators on PA in COPD patients [31,39,40,41,42,43,44,45,46,47,48,49,50,51,52].

In this review, we listed the indicators of PA and the factors influencing the reproducibility of indicators of PA, and reviewed reports that evaluated the effects of bronchodilators on PA in COPD patients using an accelerometer. Then, we investigated the influence of these factors on the results of the effectiveness of bronchodilator treatment. Finally, we described the sedentary behavior that has received attention as a risk factor for mortality independent of PA.

## 2. Indicators of PA

Several indicators have been used to evaluate PA with an accelerometer. Accelerometers are roughly divided into two types: one is a device that detects the duration of kinds of activity; the other detects the duration of the intensity of activity. Both types of accelerometers can measure total PA and the daily step count. An association map of each indicator is shown in Figure 2. The intensity levels are expressed as METs (the ratio of the working metabolic rate to the standard resting metabolic rate), and 1.0 MET is considered to be the resting metabolic rate obtained during quiet sitting. Activity with an intensity of ≥6.0 METs is classified as VPA, ≥3.0 METs to <6.0 METs is classified as moderate-intensity PA (MPA), >1.5 METs to <3.0 METs is classified as light-intensity PA (LPA), and ≤1.5 METs (usually ≥1.0 METs) is classified as sedentary behavior [53].

### 2.1. Duration of MVPA

The duration of MVPA, defined as an intensity of ≥3.0 METs, is one of the most frequently employed indicators of PA [54,55,56,57,58,59]. Three METs is considered the metabolic rate obtained walking at 4.0 km/hour on flat and firm surfaced road, which is thought to be the normal walking speed of adults [60]. The American College of Sports Medicine and the American Heart Association recommend moderate-intensity aerobic PA for a minimum of 30 min on five days per week for all adults of ≥65 years of age or adults of 50 to 64 years of age who have a chronic condition or functional limitation in order to promote and maintain health [61]. However, there is no recommendation for COPD at present.

### 2.2. Duration of LPA with MVPA

In some reports, the duration of LPA+MVPA is used as an indicator of PA because LPA is “bodily movement produced by skeletal muscles” [18]. However, LPA is “an insufficient physical activity level to meet present physical activity recommendations” and is included in the definition of “physical inactivity” [20]. Thus, when LPA+MVPA is employed as an indicator, investigators should recognize that it partly includes the duration of physical inactivity.

### 2.3. Duration of Walking and/or Standing

This indicator is obtained when an accelerometer that detects the duration of kinds of activity is employed. In the first report about PA in patients with COPD that evaluated PA using an accelerometer, which was reported by Pitta et al. [62], the durations of walking and standing were used as indicators of PA. They reported that the percentages of walking and standing time were 6% and 27%, respectively in COPD patients, and 11% and 41% in healthy subjects [62]. Not taking into account the intensity of PA might be a weakness of these indicators.

### 2.4. Total Activity

This is a frequently used indicator of PA. Total activity is defined as the total daily energy expenditure in kcal or the accumulated MET value multiplied by the hours of overall activity or the activity defined according to the MET established thresholds (usually 3.0, 2.0, or 1.5 METs) [63]. The total activity divided by resting energy expenditure or the average movement intensity during walking are also called the “physical activity level” [64,65] or the “movement intensity” [62].

### 2.5. Step Count

The daily step count is one of the indicators of PA [57]. It is more familiar and easily understandable for the general population than other indicators. However, it does not include the factor of intensity. 

### 2.6. Sedentary Time

As an accelerometer cannot detect PA of <1.0 METs, the duration of PA of ≤1.5 METs and ≥1.0 METs in intensity is used as a practical indicator of sedentary time [22], although sedentary time is defined as waking time in which the intensity of PA is ≤1.5 METs [20,23]. Although activity of 1.0 to 1.5 METs can sometimes be recorded during sleep, the sleeping time should be excluded from the sedentary time. Thus, investigators should instruct patients to attach an accelerometer from the time they wake up until the time they go to sleep or should select the data obtained during the subject’s waking time from all measured data. Even after this type of processing, the exclusion of nap times during the day is difficult.

## 3. Factors Influencing the Reproducibility of Data

When investigators evaluate the effects of some interventions, they should minimize the effects of other factors that may influence the reproducibility of the PA data.

### 3.1. Non-Wear Time

For the measurement of PA with an accelerometer, investigators usually instruct subjects to wear an accelerometer for a specified period of time in the day (e.g., 24 h, from waking up until going to bed, or from 8:00 a.m. to 9:00 p.m., etc.) except for bathing time and water activities. However, subjects sometimes forget to attach the accelerometer, especially when they wake up, after bathing, or after changing their clothes. When it is not worn, the accelerometer would show a value of 0 METs, even when the subject is active. Although the use of an accelerometer is considered a more objective and reliable method for evaluating PA than a questionnaire, the contamination of data by the non-wearing of the accelerometer is one of the limitations of evaluating PA using an accelerometer. Byron et al. [63] reported that the majority of articles did not report how non-wear time was identified (67 of 76 studies, 88%).

#### 3.1.1. Definition of Non-Wear Time

Different definitions of wearing time may alter the average accelerometer counts [66]; thus, the definition of non-wear time is important. In some studies, a 60-min period of non-measurement was defined as evidence of the device not being worn [67]. Recently, a more precise definition of non-wear (90 min of consecutive non-measurement with allowance for 2 min of interruptions) was used for patients with COPD [68,69].

#### 3.1.2. Minimum Required Wearing Time

The determination of minimum required wearing time is as important for eliminating invalid days as the non-wear time. Byron et al. [63] reported that most studies did not report how a valid day of device wearing was determined (36 of 76 studies, 47%). Among studies that did report definitions of a valid day of device wearing, most of the studies set the minimum required wear time as 12 h (15%), 10 h (16%), or 8 h (9%).

### 3.2. Days of Special Behavior

#### 3.2.1. Days with Uncommon Activities

To obtain reproducible data, the data of the days with uncommon activities (e.g., travel, events with exercise, days of sickness, etc.) should be excluded from the analysis. To detect these days, investigators should instruct subjects to record their uncommon activities in a diary.

#### 3.2.2. First and Last Days of Measurement

Subjects visit hospital to attach an accelerometer on the first day and to remove it on the last day; thus, the measurements on these days are incomplete. Investigators should therefore exclude the data of these days from the analysis.

### 3.3. Environmental Factors

#### 3.3.1. Weather

PA is largely affected by weather. On rainy days, PA is significantly reduced in comparison to non-rainy days [70,71,72,73]. The time spent performing PA of ≥2.0 METs and ≥3.0 METs were 145.6 min and 21.3 min, respectively, on non-rainy days and 92.8 min and 11.1 min on rainy days [70]. The daily step count was 3999 on dry days and 3771 on rainy days [71]. Rainfall of 10 mm translated to a decrease of approximately 175 steps [74]. Furthermore, the questionnaire-evaluated difficulty scores of the proactive physical activity in COPD on heavy rainfall days were significantly higher in comparison to light to moderate rainfall days [73]. Therefore, the data of rainy days should be excluded from analyses to ensure the reproducibility of data obtained before and after intervention.

#### 3.3.2. Season

PA is affected by season, especially by the air temperature on the day of monitoring. The activity count in summer (*n* = 23) was significantly longer than that in winter (*n* = 20) in English COPD patients (*p* = 0.01) [75]. The duration of PA of ≥2.0 METs in intensity in summer was significantly longer than that in winter in both Brazilian and Belgian COPD patients, although the activity level in Brazil was always higher than that in Belgium [76]. These differences in seasons might be caused by the difference in air temperature in each season. Donaldson et al. [77] reported that when the average daytime and nighttime temperature was ≤20.5 °C, more patients went out as the temperatures became warmer; the OR was 1.028 per 1 °C rise in temperature in English COPD patients. At <2.5 °C, the increase in patients going outdoors with temperature became significantly faster, with the OR at 1.13 per 1 °C rise in temperature. At >20.5 °C, patients reduced outdoor activity, with an OR at 0.96 per 1 °C rise. Alahmari et al. [71] reported that when the average temperature was ≤22.5 °C, the daily step count increased 43 steps per 1 °C rise in English COPD patients. When the temperature was >22.5 °C, the daily step count fell by −891 steps per 1 °C increase in temperature. Vaidya et al. [73] reported that although there was a tendency for the questionnaire-evaluated PA to decrease with higher temperatures, the seasons did not have any influence on the PA score in French COPD patients. Balish et al. [74] reported that daily step count increased 316 steps for each 10 °C rise in temperature in Canadian COPD patients. The complete elimination of the effects of season is difficult, however, careful data management is required to obtain reproducible data when PA is monitored in extremely hot or extremely cold seasons.

#### 3.3.3. Air Pollution

Some investigators reported that PA is affected by air pollution, but others did not. Alahmari et al. [71] reported that time spent outdoors fell with higher ozone levels but not with PM10. After controlling for climatic and other variables, pollutant levels of >60–70 μg/m^3^ influenced both the time outdoors and daily step count. On the other hand, Vaidya et al. [73] reported that the questionnaire-evaluated PA was not correlated with the condition of main atmospheric pollutants (ATMO index: the level of particulate matter <10 µm in size, including PM10, ozone, nitrogen dioxide, and sulfur dioxide). The effects of air pollution on PA are still controversial.

#### 3.3.4. Holidays

Holidays might affect PA. In healthy subjects, the PA on weekend days was significantly different from that on weekdays [78,79], and some investigators recommended the inclusion of data from both weekdays and weekend days [80]. In COPD patients, however, the PA on holidays did not differ from that on weekdays [72,81]. This difference might be attributed to the fact that most healthy subjects were working, while most COPD patients were retired. When PA is investigated in retired COPD patients, investigators might not need to account for holidays.

### 3.4. Number of Valid Days Required

After excluding the days of non-wear, the days of special behavior, and the days with environmental factors, it is important that the number of remaining days (i.e., valid days) is sufficient to obtain reproducible data. However, more than two-thirds of reports did not describe the minimum number of valid days required. Of the reports that did describe this information, the number ranged from 2 to 7 days [63]. Watz et al. [65] reported that a period of 2 to 3 days was sufficient to obtain a reliable measurement of physical activity in GOLD stage IV patients, whereas up to 5 days of measurement was required for patients with GOLD stage I. When we evaluated the minimum number of days required after excluding rainy days and days of special behavior, at least a 3-day measurement was required to obtain reproducibility for both the Actimarker™ (triaxial accelerometer; Panasonic, Osaka, Japan) [70] and the Active Style Pro HJA-750C™ (triaxial accelerometer; Omron Healthcare, Kyoto, Japan) [72].

## 4. Effects of Bronchodilators on Physical Activity

### 4.1. Methods

In May 2020, we searched the PubMed database for original articles written in the English language, published from 2005 to 2020 that were related to the effects of bronchodilators on PA in COPD patients. The following search term was used: “COPD”, “bronchodilator(s)”, “physical activity”, or “sedentary”. 

### 4.2. Effectiveness in COPD

Among a total of 392 reports related to the effects of bronchodilators on PA in COPD patients that were registered in the PubMed database from 2005 to 2020, 15 reports that measured PA with an accelerometer were selected (Figure 3) [31,39,40,41,42,43,44,45,46,47,48,49,50,51,52]. Among these 15 reports, 8 studies reported that bronchodilators were effective for improving PA, 3 studies reported that some bronchodilators were suggested to be effective by some indicators but not other indicators, and 4 studies reported that bronchodilators were ineffective (Table 2, Table 3 and Table 4).

#### 4.2.1. Studies Reporting that Bronchodilators Were Effective

Watz et al. [39] compared the effects of three-week treatment with indacaterol or placebo in 129 patients with moderate to severe COPD (FEV1 % predicted: 64%, age 61.4 years) using a crossover design. The MVPA, total activity (PA level), and step count were used as indicators. Over one week of monitoring with a biaxial accelerometer (SenseWear^®^ Armband^®^; BodyMedia, Pittsburgh, PA, USA), the data of non-wear days (<22 h/day) and cases with less than three valid days were excluded from the analysis. The MVPA, PA level, and step count in the indacaterol group increased by 10.3 min, 1.61 times, and 722.4 steps, respectively, in comparison to the placebo group.

Watz et al. [40] compared the effects of four-week combination treatment with aclidinium/formoterol in 127 patients with moderate to severe COPD (FEV1 % predicted: 60.3%, age 62.6 years) to placebo in 123 patients with moderate to severe COPD (FEV1 % predicted: 61.0%, age 62.1 years) using a parallel group design. The indicators employed were MVPA, total activity (PA level), and step count. In one week of monitoring with a triaxial accelerometer (DynaPort MoveMonitor^®^; McRoberts B.V., the Hague, the Netherlands), the data of non-wear days (<8 h/day) and cases with <3 valid days were excluded from the analysis. The MVPA, total activity, and step count in the aclidinium/formoterol group increased by 9.7 min, 40.9 kcal, and 731 steps, respectively, in comparison to the placebo group.

We examined the effect of six-week treatment with tiotropium/olodaterol combination or tiotropium in 184 patients with moderate to very severe COPD (FEV1 % predicted: 52.6%, age 72.8 years) in a post hoc analysis of a crossover study. The indicators employed were MVPA and sedentary time. In two-weeks of monitoring with a triaxial accelerometer (Active Style Pro HJA-750C^®^; Omron Healthcare, Kyoto, Japan), the data of non-wear days (<10 h/day) and days with environmental factors (rainy days), and cases with <3 valid days were excluded from the analysis. In the tiotropium/olodaterol group, The MVPA increased by 2.6 min and sedentary time decreased by 8.64 min in comparison to the tiotropium group [41].

Five other reports employed an observational design with a small number of patients. Hataji et al. [42] examined the effects of a four-week treatment with indacaterol in 23 patients with mild to very severe COPD (FEV1 % predicted: 64.5%, age 69.7 years). The indicators employed were MVPA, total activity, and step count. Over four weeks of monitoring with a uniaxial accelerometer (Lifecorder^®^; Suzuken, Nagoya, Japan), invalid data processing was not employed. The MVPA, PA level, and step count increased by 10.3 min, 45.2 kcal, and 1616 steps, respectively.

We examined the effect of six-week treatment with bronchodilators in 21 patients with mild to very severe COPD (FEV1 % predicted: 52.6%, age 70.7 years). The indicators employed were MVPA and total activity. In two-week monitoring with a triaxial accelerometer (Active Style Pro HJA-750C^®^), the data of special behavior days (first and last days) and day with environmental factors (rainy days), and cases with <3 valid days were excluded from the analysis. The MVPA and total activity increased by 6.8 min and 0.45 METshour, respectively [43].

Kamei et al. [44] compared the effects of 8-week treatment with aclidinium in 22 patients with moderate to severe COPD (FEV1 % predicted: 60.1%, age 72.3 years) or tiotropium in 22 patients with moderate to severe COPD (meanFEV1 % predicted: 57.6%, age 70.9 years) using a parallel group design. The indicator employed was sedentary time. In one-week of monitoring with a triaxial accelerometer (GT3X-BT^®^, ActiGraph, Pensacola, FL, USA), invalid data processing was not employed. There was no difference in sedentary time between the two treatment groups; however, the sedentary time was significantly decreased by eight-week treatment with both drugs (aclidinium 55.46 min, tiotropium 35.18 min) in comparison to the baseline values.

Hirano et al. [45] examined the effect of eight-week treatment with the assisted use of procaterol, a short acting beta-2 agonist, in 14 patients with moderate to very severe COPD (FEV1 % predicted: 55.6%, age 72.1 years). The indicators employed were MVPA and total activity. In the two-week monitoring period with a triaxial accelerometer (Active Style Pro HJA-750C^®^), the data of special behavior days (first and last days) and days with environmental factors (rainy day, holiday), and cases with <3 valid days were excluded from the analysis. The MVPA and total activity increased by 18.4 min and 1.11 METshour, respectively.

Tsujimura et al. [46] examined the effect of 12-week treatment with the assisted use of procaterol in 12 patients with severe to very severe COPD (FEV1/FVC: 34.5%, age 71.5 years). The indicators employed were total activity and step count. In the two-week monitoring period with a uniaxial accelerometer (Lifecorder^®^), invalid data processing was not employed. Total activity increased by 23.9 min and 31.8 min at 4 and 12 weeks, respectively, and the step count increased by 1021 steps and 1318 steps at 4 and 12 weeks, respectively.

#### 4.2.2. Studies in Which the Success of Bronchodilators Depended on the Indicator

In three reports, bronchodilators had a beneficial effect on PA with some indicators but not with other indicators. 

Beeh et al. [31] examined the effect of 3-week treatment with aclidinium or placebo in 112 patients with moderate to severe COPD (FEV1 % predicted: 56.7%, age 60.3 years) using a crossover design. The indicators employed were MVPA, total activity, PA level, and step count. In one week of monitoring with a biaxial accelerometer (SenseWear Pro3^®^; BodyMedia, Pittsburgh, PA, USA), invalid data processing was not employed. The MVPA and total activity in the aclidinium group increased by 10.1 min and 54.5 kcal, respectively, in comparison to the placebo group. However, the PA level and step count did not increase.

Watz et al. [47] examined the effect of three-week treatment with indacaterol/glycopyrronium in 194 patients with mild to very severe COPD (FEV1 % predicted: 61.6%, age 62.8 years) using a crossover design. The indicators employed were MVPA, total activity, PA level, and step count. In a one-week monitoring period with a biaxial accelerometer (SenseWear^®^ Armband^®^) the data of non-wear days (<21.5 h/day) and special behavior days were excluded from the analysis. Total activity, PA level, and step count in the indacaterol/glycopyrronium group increased by 36.7 kcal, 0.02, and 358 steps, respectively, in comparison to the placebo group. However, the MVPA did not increase.

Nishijima et al. [48] examined the effect of 8-week treatment with indacaterol in 18 patients with moderate to very severe COPD (FEV1 % predicted: 55.2%, age 74.2 years) with an observational design. The indicators employed were MVPA, total activity, and step count. In one week of monitoring with a uniaxial accelerometer (Lifecorder^®^), the data of days with environmental factors (rainy days, holidays), and cases with <3 valid days were excluded. The step count in indacaterol group increased by 350 steps in comparison to the placebo group. However, the MVPA and total activity did not increase.

#### 4.2.3. Studies Reporting that Bronchodilators Were Ineffective

O’Donnell et al. [49] examined the effect of 3-week treatment with indacaterol or placebo in 89 patients with mild to severe COPD (FEV1 % predicted: 61.2%, age 62.8 years) using a crossover design. The indicators employed were MVPA and total activity. In 5-day monitoring with a biaxial accelerometer (SenseWear^®^ Armband^®^), invalid data processing was not employed. In the indacaterol group, neither MVPA nor total activity increased in comparison to the placebo group. 

Troosters et al. [50] examined the effect of 24-week treatment with tiotropium in 238 patients with moderate COPD (FEV1 % predicted: 65.6%, age 61.2 years) or placebo in 219 patients with moderate COPD (FEV1 % predicted: 65.8%, age 62.3 years) using a parallel group design. The indicators employed were MVPA and step count. In a one-week monitoring period with a biaxial accelerometer (SenseWear^®^ Armband^®^), invalid data processing was not employed. In the tiotropium group, neither MVPA nor step count increased in comparison to the placebo group. 

Ichinose et al. [51] examined the effects of six-week treatment with tiotropium/olodaterol combination or tiotropium in 184 patients with moderate to very severe COPD (FEV1 % predicted: 52.6%, age 72.8 years) using a crossover design. The indicators employed were MVPA, total activity, and step count. In two weeks of monitoring with a triaxial accelerometer (Active Style Pro HJA-750C^®^), invalid data processing was not employed. In the tiotropium/olodaterol group no indicators increased in comparison to the tiotropium group. 

Troosters et al. [52] examined the effects of 12-week treatment with tiotropium/olodaterol in 65 patients with moderate to very severe COPD (FEV1 % predicted: 56%, age 64.2 years), tiotropium in 67 patients with moderate to very severe COPD (FEV1 % predicted: 57%, age 65.4 years), or placebo in 219 patients with mild to very severe COPD (FEV1 % predicted: 59%, age 64.9 years) using a parallel group design. The indicator employed was the step count. In one week of monitoring with a triaxial accelerometer (DynaPort MoveMonitor^®^), invalid data processing was not employed. The step counts in the tiotropium/olodaterol group and tiotropium group did not increase in comparison to the placebo group. 

### 4.3. Effectiveness and Influencing Factors

Among the eight studies that reported that bronchodilators were effective, a crossover design was employed in two studies, a parallel group design was employed in one, and an observational design was employed in five. Among the three studies in which effectiveness was reported to be dependent on indicators, a crossover design was employed in two studies and an observational design was employed in one. Among the four studies that reported that bronchodilators were ineffective, a crossover design was employed in two studies and a parallel group design was employed in two. The mean age of the participants in the studies that reported that bronchodilators were effective ranged from 61.4 to 72.8 years, that in studies that reported that effectiveness depended on the indicator ranged from 60.3 to 74.2 years, and that in studies that reported that bronchodilators were ineffective ranged from 61.2 to 72.8 years. The mean FEV1 % predicted values in the studies that reported that bronchodilators were effective ranged from 52.6% to 64.5% (in one study, the value of FEV1 % predicted was not described but FEV1/FVC was 34.5%), that in reports in which effectiveness was dependent on indicators ranged from 55.2% to 61.6%, and that in studies that reported that bronchodilators were ineffective ranged from 52.6% to 65.6%. None of these factors seemed to differ according to the effectiveness of bronchodilators; however, the number of observational studies was high.

Regarding factors that could affect the reproducibility of the accelerometric data, adjustment was performed for at least two of four influencing factors (non-wear time, days of special behavior, environmental factors, number of valid days required) for data processing in five of eight studies that reported that bronchodilators were effective, two of three studies that reported that the efficacy of bronchodilators depended on indicators, and none of the four studies that reported that bronchodilators were ineffective. Furthermore, in all reports that adjusted for at least two influencing factors, bronchodilators were found to have a beneficial effect on PA with some indicators (Table 4).

### 4.4. Possible Characteristics of Patients in Whom Bronchodilators Are Effective

In the post hoc analysis of the crossover study with tiotropium/olodaterol and tiotropium [41], the change in the duration of MVPA with tiotropium/olodaterol treatment was significantly larger in patients with a better pulmonary functions, less dyspnea, a longer duration of MVPA, and a shorter sedentary time at baseline in comparison to that with tiotropium. Similar results were reported by Hirano et al. [45], who noted that the changes in the total PA level with eight-week treatment with the assisted use of procaterol, a short acting beta-2 agonist, were positively correlated with both baseline FVC % predicted and FEV1 % predicted. 

## 5. Sedentary Time as a New Indicator

### 5.1. Importance of Sedentary Time for Other Conditions

Recently, the importance of sedentary time for healthcare in the general population [82,83,84,85,86,87] or patients with cardiovascular disease [88,89,90], diabetes mellitus [89,91,92], and cancer [83,93,94,95,96] has become receiving a great deal of attention. Reducing sedentary time can improve both health conditions and the mortality [97]. Interestingly, the interventions focusing on lifestyle or sedentary behavior resulted in the greatest reduction in sedentary time but those focusing on MVPA or MVPA with sedentary behavior did not [98]. Bakrania et al. [99] proposed that behavior conditions should be divided into four mutually exclusive behavioral categories according to the statuses of PA and sedentary: ‘Busy Bees’, physically active with low sedentary status; ‘Sedentary Exercisers’, physically active with high sedentary status; ‘Light Movers’, physically inactive with low sedentary status; and ‘Couch Potatoes’, physically inactive with high sedentary status. These categories better reflected the cardiometabolic health conditions.

### 5.2. Importance of Sedentary Time in Patients with COPD

The importance of sedentary behavior in patients with COPD has been reported. Ukawa et al. [100] reported that male patients with COPD who watched ≥4 h/day of television were more likely to die of COPD than those watching <2 h/day (hazard ratio 1.63; 95% confidence interval, 1.04–2.55), independent of major confounders. Furlanetto et al. [101] reported that sedentary behavior was an independent predictor of mortality after adjusting for MVPA and several other variables, including sex, age, body mass index, educational level, FEV1 % predicted, and 6-min walk distance when 101 cases of COPD were followed for 120 months (Figure 4). The strongest independent cutoff value for predicting mortality was ≥8.5 h/day of sedentary time. 

Donaire-Gonzalez et al. [102] reported that a higher level of both low- and high-intensity PA reduced the risk of COPD hospitalization in patients with FEV1 % predicted values of ≥50%. In patients with FEV1 % predicted values of <50%, however, a higher level of low-intensity physical activity reduced the risk of hospitalization, but high-intensity physical activity did not. McKeough et al. [103] reported that there was a reduced risk of all-cause mortality in “Busy Bees” in comparison to “Couch Potatoes” (hazard ratio: 0.26, 95% CI: 0.11, 0.65) after adjustment for age, gender, COPD severity, history of cardiovascular disease, history of cancer, history of diabetes, self-reported longstanding illness, body mass index, smoking status, age at completion of full-time education, and alcohol consumption. Dogra et al. [104] reported that replacing 30 min of sitting time per day with 30 min of light to moderate-intensity PA or VPA per day led to significant improvements in FEV1 % predicted among adults with an obstructive lung disease based on an isotemporal substitution analysis (Figure 5).

### 5.3. Effect of Bronchodilators on Sedentary Time

Only two post hoc or observational studies reported on sedentary time. Kamei et al. [44] reported that eight weeks of treatment with tiotropium and aclidinium reduced sedentary time by 55.46 min (95% CI −98.15, −12.77) and 35.18 min (95% CI −67.41, −2.94), respectively. We reported that the six-week treatment with tiotropium/olodaterol resulted in a significantly reduced sedentary time in comparison to tiotropium (mean difference: −8.6 min, 95% CI: −16.88, −0.40) [41]. 

## 6. Discussion

The duration of MVPA, duration of walking and/or standing, total activity, and step count were frequently employed as indicators of PA in studies in which PA was evaluated by an accelerometer. Among the previous reports that evaluated the effect of bronchodilators on PA in COPD patients, adjustment was performed for at least two of four influencing factors (i.e., non-wear time, data of days of special behavior, environmental factors, and number of valid days required) for data processing in most of the studies reporting that bronchodilators were effective or that the effectiveness was dependent on the indicator, while none of the studies reporting that bronchodilators were ineffective performed such adjustment. Furthermore, in all reports that adjusted for at least two influencing factors, bronchodilators showed beneficial effects on PA with some indicators. Thus, the processing of influencing factors to secure the reproducibility of the data might affect the results regarding the effectiveness of bronchodilators. Sedentary time is another indicator of the physical condition of COPD patients and predicts mortality independently of PA. Reducing the sedentary time could become an important target for the management of COPD.

Originally, PA was defined as “any bodily movement produced by skeletal muscles that results in energy expenditure” [18]; thus, it should include some condition of physical inactivity (e.g., LPA+MVPA). However, PA is usually defined as a physically active condition in which the MVPA exceeds a certain threshold (in minutes) [19]. Thus, the duration of MVPA and the total activity >3.0 METs [63] are frequently used as indicators. The step count is a familiar indicator of PA for the general population, but it does not reflect the intensity of the PA. Different walking speeds will produce a different total level of activity, even when the number of steps is the same. Investigators should recognize that the difference in indicators might reflect different aspects of PA. Even when the same indicator (e.g., MVPA, total activity, or step count) was employed in different studies of COPD patients, the effectiveness of bronchodilators in improving PA were inconsistent. Thus, the influence of indicators on the inconsistency of the effectiveness of bronchodilators seems to be limited.

In contrast to the beneficial effects of bronchodilators on the pulmonary function, dyspnea, exercise capacity, and exacerbations in patients with COPD, the effect on PA remains controversial. The inconsistent effects on PA might be due not only to factors that can directly affect PA (e.g., psychological conditions, comorbidities, hobbies, lifestyle, etc.) but also to factors that could affect the reproducibility of the accelerometric data (e.g., non-wear time, days of special behavior, environmental factors, and number of valid days required). We compared the effects of bronchodilators on PA under several study conditions (Table 2). 

Adjustment for at least two influencing factors was performed during data processing in five of eight studies that reported that bronchodilators were effective, two of three studies that reported that the effectiveness of bronchodilators depended on indicators, and none of the four studies that reported that bronchodilators were ineffective. Furthermore, in all reports that performed adjustment for at least two influencing factors, bronchodilators showed a beneficial effect on PA with some indicators. There were clear differences among the three effectiveness groups. Thus, the processing of factors influencing reproducibility might be one of the reasons for the different effectiveness of bronchodilator. In one of three studies that reported that bronchodilators were effective in which there was no adjustment for reproducible factors, the duration of monitoring was longer (four weeks) in comparison to the other 14 reports (between five days and two weeks). This longer duration of monitoring might have improved the reproducibility of the data. From these differences, the effectiveness of bronchodilators might largely depend on the reproducibility of data. In our crossover study, the duration of MVPA did not different between tiotropium/olodaterol and tiotropium when all data obtained from two weeks of monitoring were included [51]. However, in post-hoc study, the duration of MVPA was significantly improved with tiotropium/olodaterol in comparison to tiotropium when adjustment was performed for the factors of non-wear time, days of special behavior, weather, and number of valid days [41]. Thus, investigators should carefully process the monitoring data to obtain reproducibility when evaluating the effectiveness of intervention.

On the other hand, the effectiveness of bronchodilators did not seem to differ according to background factors, including study design, age, and the lung function, although the studies that reported that bronchodilators were effective included a relatively large number of observational studies. 

Several factors, including psychological conditions (e.g., anxiety and motivation), comorbidities (e.g., anemia, diabetes mellitus, osteoporosis, depression, or cardiovascular disease), hobbies, lifestyle, and other factors might affect PA in COPD patients. The influence of the fluctuation of these factors on PA might be greater than the influence of bronchodilators, even if the reproducibility of the data was secured. Hobbies and lifestyle were relatively stable before and after intervention; however, the psychological condition and condition of comorbidities might relatively easily fluctuate. The description of these conditions could not be found in the reports selected in the current review. It might be important to maintain consistency in these factors when evaluating the effects of bronchodilators on PA in future studies. Under such circumstances, more reproducible data are required for evaluating the physical activity, especially when an accelerometer is used. Therefore, adjusting for the non-wear time, days with special behavior, environmental factors, and number of valid days should be conducted.

The changes in the duration of MVPA of patients treated with bronchodilators was significantly larger in the patients who had a better pulmonary function, less dyspnea, and better PA at baseline [41,45]. These results suggest that early intervention with bronchodilators might provide a better improvement of PA in patients with COPD and might maintain a more active lifestyle in comparison to late intervention, although these results should be interpreted with caution because they were obtained from two post hoc or observational studies. Further studies are required to clarify these relationships.

The importance of reducing sedentary time has also been emphasized [53] because sedentary behavior was a risk factor for mortality, independent of MVPA [101]. Thus, physicians should measure and treat both PA and sedentary behavior to improve the health outcomes of patients with COPD [105,106,107]. Furthermore, in patients with mild to moderate COPD, the risk of hospitalization decreased as the step count increased, regardless of the level of total activity. In patients with severe to very severe COPD, however, the risk of hospitalization decreased as the step count increased when their total activities were low-intensity, but the risk increased as the step count increased when their total activities were high-intensity [102]. Sedentary behavior should be changed to LPA and MVPA in cases of mild to moderate COPD, but it should be changed to LPA but not to MVPA in cases of severe to very severe COPD. Effects of bronchodilators on sedentary time were observed in two reports; however, they were post hoc or observational studies. Further studies with adjustment to improve reproducibility are required to clarify the effect of bronchodilators on the sedentary time.

The current review is associated with several limitations. First, the number of reports that evaluated the effect of bronchodilators on objectively measured PA was relatively small, especially with regard to studies with high-level evidence. The accumulation of further studies with high levels of evidence is necessary. Second, the type, pharmacological actions, and durations of action times of the bronchodilators that were evaluated were heterogeneous. The effectiveness of bronchodilators on PA might differ among these agents, however, the number of reports was not sufficient to clarify these relationships. Third, it is unclear whether the differences that were induced by bronchodilators in the studies that were reviewed reached a minimal clinical important difference (MCID) in studies that reported that bronchodilators were effective. The MCID of PA in patients treated with bronchodilators has not been demonstrated, although the MCID in step count in patients who undergo rehabilitation has been reported to be 600 to 1100 steps [108]. Further studies are required to confirm the MCID.

## 7. Conclusions

Although the effect of bronchodilators on PA in patients with COPD was controversial, all of the reports that included adjustment for factors influencing the reproducibility demonstrated beneficial or partially beneficial effects. Further studies to evaluate the effect of intervention on PA should incorporate data processing in consideration of reproducibility.

## Figures and Tables

**Figure 1 jcm-09-03497-f001:**
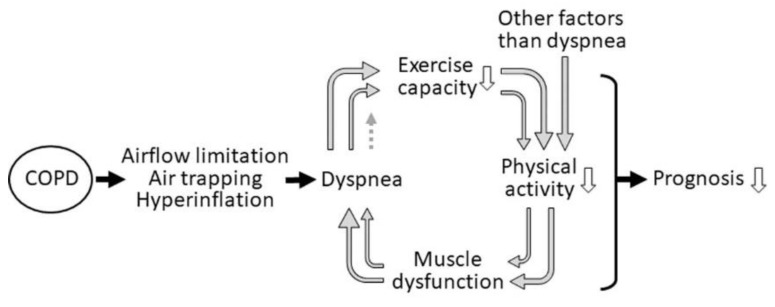
The downward spiral of inactivity in COPD. Physiological changes lead to dyspnea and decreased exercise capacity. Physical activity is reduced by decreased exercise capacity and other factors, such as psychological, socio-economic, comorbidities, and/or skeletal muscle weakness.

**Figure 2 jcm-09-03497-f002:**
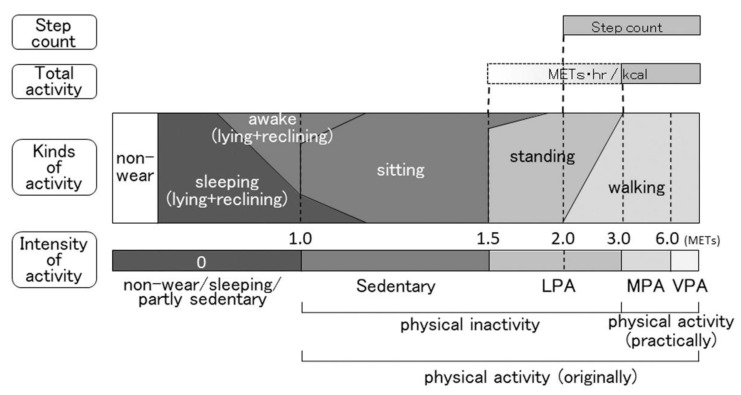
Indicators of physical activity based on the intensity of activity. The kinds of activity do not clearly correspond to the intensity of activity. Abbreviations: METs, metabolic equivalents; LPA, light-intensity physical activity; MPA, moderate-intensity physical activity; VPA, vigorous-intensity physical activity.

**Figure 3 jcm-09-03497-f003:**
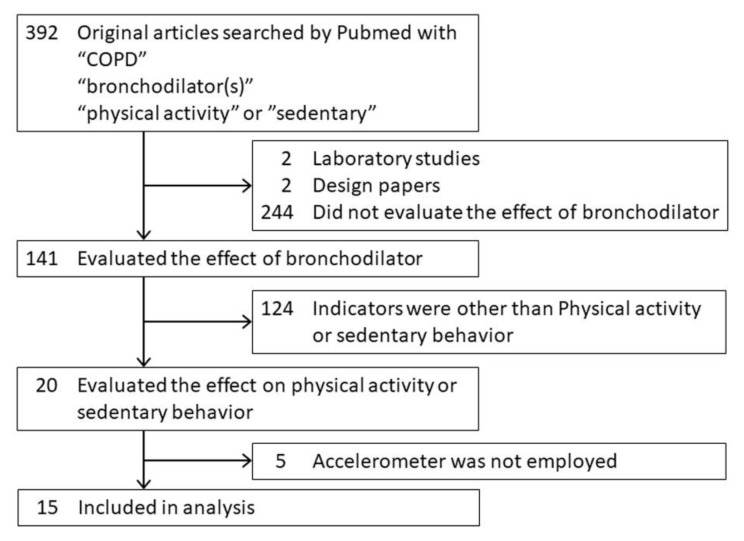
PRISMA flow diagram.

**Figure 4 jcm-09-03497-f004:**
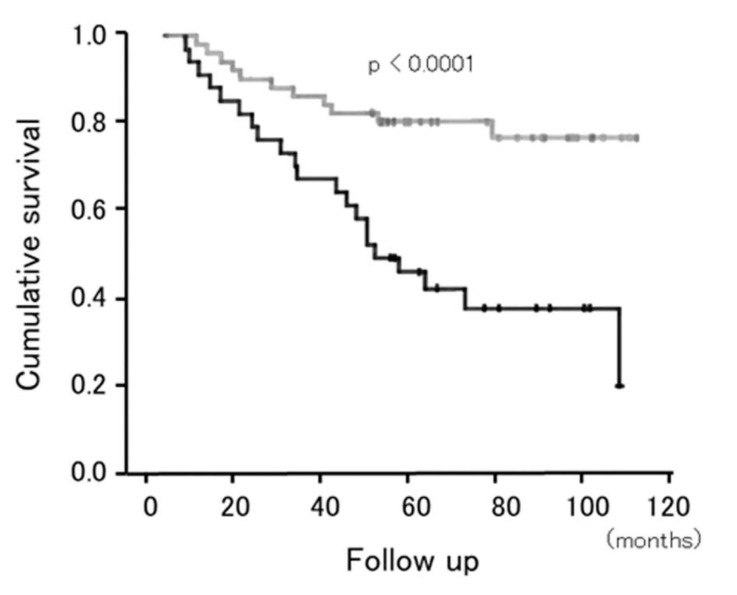
Cumulative survival according to the sedentary time. Survival rate in patients with COPD after adjusted by sex, age, body mass index, educational level, lung function, functional exercise capacity, and moderate to vigorous-intensity physical activity. Gray line indicates the patients with <8.5 h/day of time spent in sedentary activities. Black line indicates the patients with ≥8.5 h/day of time spent in sedentary activities. Adopted from [101].

**Figure 5 jcm-09-03497-f005:**
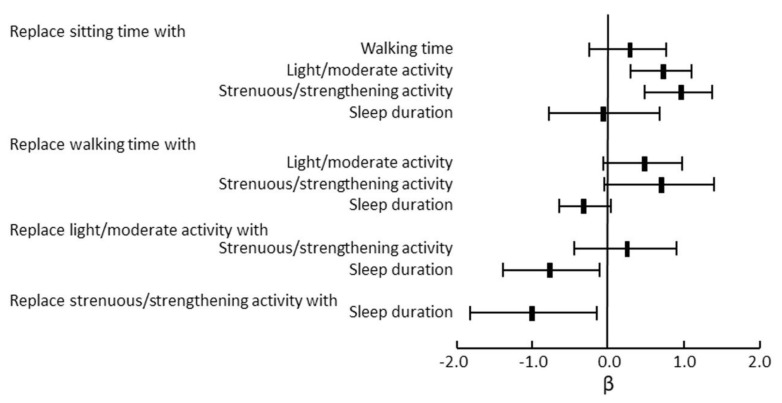
Isotemporal substitution models for FEV1 % predicted in response to 30 min per day substitution of movement behaviors in adults with obstructive lung disease. Replacing 30 min of sitting time with 30 min of light/moderate activity or strenuous/strengthening activity leads to significant improvements in FEV1 % predicted. Adopted from reference 104.

**Table 1 jcm-09-03497-t001:** Definitions of physical activity.

Terms	Definitions
Physical activity	(Originally) Any bodily movement produced by skeletal muscles that results in energy expenditure [18]
	(Practically) Physically active behavior that is comparable to MVPA [19,20]
Physical inactivity	A PA level that is not sufficient for meeting the present PA recommendations [20,21]
	Threshold: [20] The non-achievement of 150 min of MVPA per week or 75 min of VPA per week or an equivalent combination of moderate- and vigorous-intensity activity
Sedentary behavior	Any waking behavior characterized by an energy expenditure ≤1.5 METs, while in a sitting, reclining or lying posture [20,22,23]

MVPA, moderate to vigorous-intensity physical activity; VPA, vigorous-intensity physical activity; METs, metabolic equivalents.

**Table 2 jcm-09-03497-t002:** Methodological details of the articles.

Authors/Year of Publication	Country	Bronchodilator	Study Design	Accelerometer	
				Sensor Type	Product Name
**Effective**					
Hataji 2013 [42]	Japan	Ind	observation	uniaxial	Lifecorder
Watz 2014 [39]	Germany	Ind/Tio/Plac	crossover	biaxial	SenseWear armband
Minakata 2015 [43]	Japan	BD	observation	triaxial	Actimarker
Watz 2017 [40]	Germany	Acl/For vs. Plac	parallel groups	triaxial	DynaPort MoveMonitor
Minakata 2019 [41]	Japan	Tio/Olo vs. Tio	Crossover (post-hoc)	triaxial	Active Style Pro HJA-750C
Kamei 2019 [44]	Japan	Acl, Tio	Observation (post-hoc)	triaxial	ActiGraph GT3X-BT
Hirano 2019 [45]	Japan	Procat	observation	triaxial	Actimarker
Tsujimura 2019 [46]	Japan	Procat	observation	uniaxial	Lifecorder
**Dependent on Indicators**					
Beeh 2014 [31]	Germany	Acl vs. Plac	crossover	biaxial	SenseWear Pro3
Nishijima 2015 [48]	Japan	Ind	observation	uniaxial	Lifecorder
Watz 2016 [47]	Germany	Ind/Gly vs. Plac	crossover	biaxial	SenseWear armband
**Ineffective**					
O’Donnell 2011 [49]	Belgium, Canada, Denmark, Italy, Spain, USA	Ind vs. Plac	crossover	biaxial	SenseWear armband
Troosters 2014 [50]	Belgium, Canada, Czech Republic, Germany, Greece, Netherlands, Portugal, Ukraine, UK, USA	Tio vs. Plac	parallel groups	biaxial	SenseWear armband
Ichinose 2018 [51]	Japan	Tio/Olo vs. Tio	crossover	triaxial	Active Style Pro HJA-750C
Troosters 2018 [52]	Australia, Austria, Belgium, Canada, Denmark, Germany, New Zealand, Poland, Portugal, UK, USA	Tio/Olo vs. Tio vs. Plac	parallel groups	triaxial	DynaPort MoveMonitor

Ind, indacaterol; Tio, tiotropium; Plac, placebo; BD, bronchodilator; Acl, aclidinium; For, formoterol; Olo, olodaterol; Procat, procaterol; Gly, glycopyrronium.

**Table 3 jcm-09-03497-t003:** Patients characteristics of the articles.

Authors/Year of Publication	No. of Patients	Age	FEV1 % Pred	COPD Stage	Duration of Medication	Duration of Monitoring
**Effective**						
Hataji 2013 [42]	23	69.7	64.5	I–IV	4 W	4 W
Watz 2014 [39]	129	61.4	64	II, III	3 W	1 W
Minakata 2015 [43]	21	70.7	52.6	I–IV	6 W	2 W
Watz 2017 [40]	127 vs. 123	62.6 vs. 62.1	60.3 vs. 61.0	II, III	4 W	1 W
Minakata 2019 [41]	184	72.8	52.6	II, III, IV	6 W	2 W
Kamei 2019 [44]	22 vs. 22	72.3 vs. 70.9	60.1 vs. 57.6	II, III	8 W	1 W
Hirano 2019 [45]	14	72.1	55.6	II, III, IV	8 W	2 W
Tsujimura 2019 [46]	12	71.5	FEV1% 34.5	III, IV	4, 12 W	2 W
**Dependent on indicators**						
Beeh 2014 [31]	112	60.3	56.7	II–III	3 W	1 W
Nishijima 2015 [48]	18	74.2	55.2	II, III, IV	12 W	1 W
Watz 2016 [47]	194	62.8	61.6	I–IV	3 W	1 W
**Ineffective**						
O’Donnell 2011 [49]	89	62.8	61.2	I, II, III	3 W	5 days
Troosters 2014 [50]	238 vs. 219	61.2 vs. 62.3	65.6 vs. 65.8	II	24 W	1 W
Ichinose 2018 [51]	184	72.8	52.6	II, III, IV	6 W	2 W
Troosters 2018 [52]	65 vs. 67 vs. 72	64.2 vs. 65.4 vs. 64.9	56 vs. 57 vs. 59	I–IV	12 W	1 W

FEV1 % Pred, forced expiratory volume in one second % of predicted; W, week(s).

**Table 4 jcm-09-03497-t004:** Adjustment of influencing factors and effectiveness of bronchodilator.

Authors/Year of Publication	Processing of Invalid Data				MVPA	Total Activity	(PA Level)	Steps	Sedentary
	Non-Wear	Special Behavior	Environmental Factors	Number of Valid Days	(min) Increase	(METs·h or kcal) Increase	(/resting) Increase	(steps) Increase	(min) Decrease
**Effective**									
Hataji 2013 [42]	-	-	-	-	□	□		□	
Watz 2014 [39]	☆	-	-	☆	□		□	□	
Minakata 2015 [43]	-	☆	☆	☆	□	□			
Watz 2017 [40]	☆	-	-	☆	□	□		□	
Minakata 2019 [41]	☆	-	☆	☆	□				□
Kamei 2019 [44]	-	-	-	-					□
Hirano 2019 [45]	-	☆	☆	☆	□	□			
Tsujimura 2019 [46]	-	-	-	-		□		□	
**Dependent on indicators**									
Beeh 2014 [31]	-	-	-	-	□	□	■	■	
Nishijima 2015 [48]	-	-	☆	☆	■	■		□	
Watz 2016 [47]	☆	☆	-	-	■	□	□	□	
**Ineffective**									
O’Donnell 2011 [49]	-	-	-	-	■	■			
Troosters 2014 [50]	-	-	-	-	■			■	
Ichinose 2018 [51]	-	-	-	-	■	■		■	
Troosters 2018 [52]	-	-	-	-				■	

MVPA, moderate to vigorous-intensity physical activity; PA, physical activity; METs, metabolic equivalents; ☆, factor was adjusted; -, factor was not adjusted; □, beneficial effect was obtained; ■, beneficial effect was not obtained.

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
