# Peer review of "Data Reproducibility and Effectiveness of Bronchodilators for Improving Physical Activity in COPD Patients"

_jcm, 2020, doi:10.3390/jcm9113497_

Round 1

Reviewer 1 Report

I am happy that the authors have addressed the concerns raised in the review.

Author Response

Thank you for your kind comment. I greatly appreciate your review.

Reviewer 2 Report

Dear editor,

Sorry, I really do not know where to begin and end my comments to this manuscript. I would therefore refrain from providing detailed commentary. In my view the changes made by the authors did not improve the paper to a level that it is acceptable for publication. The title now covers even less than the previous one. The most important mistake the authors make is confusing measurement variablity in PA at the individual level and unreliability at the group level. In studies comparing groups, the first factor averages out and cannot explain between group differences in for instance studies on the effect of bronchodilator use on PA. This is given so much weight in the paper that in my view it affects the validity of the findings, hence should not reach the international literature.

Author Response

I greatly appreciate your valuable comments. I want to make use of your importanty opinion for further article making.

Reviewer 3 Report

Dear authors 

thank you for this interesting article. Although in my opinion some work has to be done this is a well structured and well written paper. Congrats on this.

I have some suggestions which might help to improve it.

Best regrads

the reviewer

In general:

Paper JCM 941203 by Minakata et al. is a review article addressing the reproducibility of the data and the effectiveness of the therapy with bronchodilators in COPD regarding physical activity (PA).

The text contains 2 parts. The first part describes the different levels of physical activity, how they are recorded and the factors that influence them. In the second part, COPD studies are reviewed in which the effects of bronchial dilatation therapy on PA are examined in this patient group.

The topic is interesting and meaningful since it is known that prognosis and health related quality of life in COPD depends on maintenance of PA. So almost every intervention should  proofed in this direction even it is with inhaled bronchodilators.

All over in the first part it is not clear how the authors select the references and how they were graded. In Line 103 they mentioned that they “summarized” the indicators and the factors but the authors missed to explain how did this process happened. This should be added. In the second part they report how they carried out the search and evaluation.

Based on the influencing factors found on PA e.g. measured simply as steps per day the authors correctly regard this outcome parameter for clinical studies as critical. They recommend regarding the effects on sedentary behavior by medical intervention as a more meaningful parameter or at least a combination of measuring PA and sedentary time. I agree with this conclusion.

Some sentences are difficult to read with an awkward wording. Anyway the paper should be reviewed by a native speaker. The text still contains some careless mistakes. Some are outlined below.

Factors influencing the data are mentioned in chapter 3. Normally interventional trials include a larger number of participants leading to a levelling of errors with an uniform distribution. This is why control groups should be included. Please comment on this.

Abstract:

This abstract has a special structure since it not provides the usual structure: methods: how many studies are were included or how many participants were observed. Also no concrete results in form of main statistical findings are mentioned. It is written in one text. Maybe the classical structure clarify the missings.

Introduction

Figure 1:

  • The figure shows the wrong order, the right one is: Dyspnea-PA-Exercise Capacity- muscle function
  • This figure is more a circle than a spiral (which would be better)

Line 70-81 :This sentence is confusing and contents 4 x used or using in one sentence. “used to be mainly used” (maybe: which are mainly used)

Line 86-94 describes the definitions of PA. It would be helpful to provide a table here to illustrate the different levels of PA

Legend Figure 2 (line 120-123): yellow marked part should be deleted since it is mentioned in the text. Please check on all the legends since they contain to much information and the information is added in the text

Line 126-133 please provide also the recommendations for COPD patients which do exist

Line 146: That the intensity … wording

Line 181: two times “change” in one sentence, maybe use “alter”

Line 477: acridinium means aclidinium

Line 479: significant reduced the sedentary time

Limitation are well addressed.

Conclusion:

First sentence is not clear. Do you mean : due to the small number of reports the effects were controversial?

References: I found no mistakes

Author Response

Point 1: All over in the first part it is not clear how the authors select the references and how they were graded. In Line 103 they mentioned that they “summarized” the indicators and the factors but the authors missed to explain how did this process happened. This should be added. In the second part they report how they carried out the search and evaluation.

Response 1: Unfortunately, we did not systematically gather references concerning the indicators and factors, instead selecting several reviews and reports regarding physical activity and organizing these points. As the word “summarized” was not appropriate, we changed the word to “listed”.

Point 2: Based on the influencing factors found on PA e.g. measured simply as steps per day the authors correctly regard this outcome parameter for clinical studies as critical. They recommend regarding the effects on sedentary behavior by medical intervention as a more meaningful parameter or at least a combination of measuring PA and sedentary time. I agree with this conclusion.

Response 2: We believe that both increasing physical activity and reducing sedentary time are important for managing COPD.

Point 3: Factors influencing the data are mentioned in chapter 3. Normally interventional trials include a larger number of participants leading to a levelling of errors with an uniform distribution. This is why control groups should be included. Please comment on this.

Response 3: Generally, I agree with you. Including more participants would help reduce the error. However, as physical activity is affected by many factors, such as socio-economic conditions, psychogenic conditions, and comorbidities, changes in the physical activity due to intervention may be a bit difficult to assess, even if perfectly reliable data were used. Under such circumstances, more reproducible data are required for evaluating the physical activity, especially when an accelerometer is used. Therefore, adjusting for the non-wear time, days with special behavior, environmental factors, and number of valid days should be conducted. We have now mentioned these points in the discussion.

Point 4: Abstract: This abstract has a special structure since it not provides the usual structure: methods: how many studies are were included or how many participants were observed. Also no concrete results in form of main statistical findings are mentioned. It is written in one text. Maybe the classical structure clarify the missings.

Response 4: As suggested, we revised the Abstract.

Point 5: Figure 1: The figure shows the wrong order, the right one is: Dyspnea-PA-Exercise Capacity- muscle function.

Response 5: “Exercise capacity” is the maximal capacity of exercise, while “physical activity” is the daily activity performed within the upper limit of capacity (“exercise capacity”) and unable to be performed over that limit. I therefore believe that the “physical activity” should be suppressed after the “exercise capacity” is suppressed. I discussed this point with Dr Richard Casuburi at the 2018 ERS international conference, and his opinion was that both pathways could be possible and that the sequence did not matter. I wanted to emphasize that a decrease in the “exercise capacity” happens first, followed by a decrease in the “physical activity”. I have therefore described the sequence as “dyspnea-exercise capacity-PA-muscle function”.

Point 6: Figure 1: This figure is more a circle than a spiral (which would be better)

Response 6: I intended to present a spiral with a two-dimensional image, not a circle. The width of the arrow gradually narrows with each rotation. I would thus like to keep this two-dimensional image.

Point 7: Line 70-81: This sentence is confusing and contents 4 x used or using in one sentence. “used to be mainly used” (maybe: which are mainly used)

Response 7: We have now divided this point into two sentences, as follows: “Recently, objective measurements using an accelerometer have been widely used to evaluate PA rather than questionnaires, which were mainly used but tended to overestimate the PA. An accelerometer can detect acceleration in one, two, or three dimensions (uniaxial-, biaxial-, triaxial-accelerometer, respectively) and can calculate the values of several indicators of PA.”

Point 8: Line 86-94 describes the definitions of PA. It would be helpful to provide a table here to illustrate the different levels of PA

Response 8: I have now added a new table (Table 1) as suggested.

Point 9: Legend Figure 2 (line 120-123): yellow marked part should be deleted since it is mentioned in the text. Please check on all the legends since they contain too much information and the information is added in the text

Response 9: I have now shortened the figure legends.

Point 10: Line 126-133 please provide also the recommendations for COPD patients which do exist

Response 10: As there is no recommendation concerning COPD, we have now added the following sentence: “However, there is no recommendation for COPD at present.”

Point 11: Line 146: That the intensity … wording

Response 11: I have now corrected the wording as follows: “Not taking the intensity of PA into account might be a weakness of these indicators”.

Point 12: Line 181: two times “change” in one sentence, maybe use “alter”

Response 12: I have now corrected the sentence as follows: “Different definitions of wearing time may alter the average accelerometer counts”.

Point 13: Line 477: acridinium means aclidinium

Response 13: As it was misspelled, I corrected it.

Point 14: Line 479: significant reduced the sedentary time

Response 14: I corrected it as suggested.

Point 15: Limitation are well addressed.

Response 15: Thank you for your comment.

Point 16: Conclusion: First sentence is not clear. Do you mean: due to the small number of reports the effects were controversial?

Response 16: I have revised this point for clarity as follows: “Although the effect of bronchodilators on PA in patients with COPD was controversial, all of the reports that included adjustment for factors influencing the reproducibility demonstrated beneficial or partially beneficial effects.”

Point 17: Conclusion: References: I found no mistakes

Response 17: Thank you for your comment.

Round 2

Reviewer 3 Report

Dear Authors,

thanks for your effort. All concerns are addressed well. In my opinion the paper reached more clarity. 

Good luck for the next steps.

The Reviewer

Author Response

Thank you for your kind comment. I greatly appreciate your review.

This manuscript is a resubmission of an earlier submission. The following is a list of the peer review reports and author responses from that submission.

Round 1

Reviewer 1 Report

Thank you for allowing me to review the manuscript entitled ‘Effectiveness of bronchodilators for improving physical activity in COPD patients’

The manuscript (MS) reviews the literature on: (1) reports on PA, (2) factors affecting the reproducibility of indicators of PA in COPD, and, (3) the effects of bronchodilators on PA in COPD.

The topic ‘effectiveness of bronchodilators for improving physical activity in COPD patients’ is certainly suitable for review in my opinion. However, there are two major issues with this manuscript as it stands that devalue its scientific value significantly.

First, the content of the MS is incongruent with the title. The content is much broader than suggested in the title. If the content of the MS were to focus on the topic as stated in the title, it would make a much greater contribution to the state of science on this topic. I therefore propose to delete paragraphs two and three entirely. These paragraphs do not contribute to the primary aim and only distract.

Second, factors affecting the stability (reliability) of the measurement of PA with accelerometers were nicely reviewed. This is relevant in relation to the main aim of the review. But, how do the authors conclude that issues related to this could have affected outcomes of studies on the effects of bronchodilators on PA? I do not understand their methodology to come to this conclusion. In the results I do not find any measures of association between factors affecting the reproducibility of PA indicators and the effectiveness of bronchodilator treatment. In general, in randomized controlled studies one would expect that the prevalence of confounders is equally distributed and does not affect the primary outcome measure. Certainly, it may give noise to the signal and thus may require the inclusion of a larger number of patients. However, the majority of the included studies on the effect of bronchodilators on PA used a form of controlled design.

Lastly, the authors came across a significant number of relevant studies. Why did not they perform a meta-analysis in addition to the review?

Reviewer 2 Report

I think that this was a worthwhile question to ask but the authors do not start answering the question about bronchodilators and PA in COPD until page 8. I think that the earlier sections should be compressed and included in the introduction. Referencing in sections is poor and figures do not have legends.

Specific comments:

Introduction

Lack of references for some sentences.

Figure 1: No detail in figure legend. One should be able to read the legend alone to understand the figure.

The last part of the introduction should be in a methods section as it is describing the methodology behind the review.

Was this a systematic review? Were the articles reviewed by more than 1 person? A lot of articles were excluded – 15 out of 392 were kept. What were the reasons for this? A prisma flow diagram might be useful here.

COPD and PA

  1. Please could the authors define what an accelerometer is and what the difference between a biaxial and triaxial accelerometer is?
  2. Some of the studies from the previous section are repeated and described again which is unnecessary.

Definitions of PA

Why is this section after the COPD section? This should be placed in the introduction. Also, there is no figure legend describing figure 2 which is very complicated and unclear to follow.

Indicators of PA

Again, it is not appropriate for this section to be after the COPD section. The COPD section would be easier to read after the authors have described what accelerometers are.

The authors do not start answering the question about bronchodilators and PA in COPD until page 8. I think that the earlier sections should be compressed and included in the introduction.

Table 1: This is essentially 3 tables and therefore these should be described separately with proper legends to allow for the interpretation of the data included in the tables.

Figure 4: This is unclear and I think the authors need to fully explain the results of this and the meaning of the data.

Discussion

There are too many sub headings in the discussion. Each paragraph does not need a separate heading.